# Sustainability Monitoring with Robotic Accounting—Integration of Financial and Environmental Farm Data

Krijn Poppe [1] , Hans Vrolijk [1,*] , Nicole de Graaf [2], Roeland van Dijk [2], Emma Dillon [3] and Trevor Donnellan [3]

[1] Wageningen Economic Research, Wageningen University and Research, 2595 BM The Hague, The Netherlands; kjpoppe@hccnet.nl

[2] SOOPS, 1021 KN Amsterdam, The Netherlands; nic@soops.nl (N.d.G.); roeland@soops.nl (R.v.D.)

[3] Rural Economy & Development Centre, Teagasc, H65 R718 Athenry, Ireland; emma.dillon@teagasc.ie (E.D.); trevor.donnellan@teagasc.ie (T.D.)

* Correspondence: hans.vrolijk@wur.nl

**Abstract:** The production of farm sustainability indicators is vital for all actors in the food chain. This paper shows how robotic accounting could assist in the monitoring and compliance of farm performance, to assess the various aspects of sustainability. We show how financial farm accounting, which is routine on most farms, can be extended to deliver a range of sustainability metrics. Using farm invoices from the Netherlands and Ireland, we show that many invoices contain volume data that can be used to calculate environmental indicators such as pesticide use, mass balances (especially needed in organic farming), material balances of N and P, energy use, antibiotics use, etc. Using a number of illustrative use cases, we show the feasibility of deriving both financial and sustainability data from invoices. Standard algorithms can be used to link the invoice data to bank payment data and code it with a chart of accounts using a simple data and process model. Linking invoices with bank data provides advantages with respect to completeness, reliability, and efficiency. We describe a software tool that provides flexible data management processes that can easily be adapted by the user to collect new data that reflect emerging environmental or social concerns. Data collectors can set up procedures in which new types of data can be acquired or new indicators calculated, avoiding the need for software reprogramming. The digitalisation of invoices, ideally in a standard (UBL) format, is a necessary step to facilitate the process described. This digital format would lead to reduced accounting costs and at the same time could also provide farmers with a dashboard of sustainability indicators. Once invoices are digitalised, accounting costs drop, the potential for errors or omissions is reduced, and the administrative burden for environmental accounting diminishes due to the low marginal cost of data management.

**Keywords:** farm accounting; robotic accounting; certification; sustainability; digitalisation

## 1. Introduction

Societal concerns about the impact of agriculture on, among other things, the environment, animal welfare, and health, have led to agricultural policies and sector initiatives to improve the sustainability performance of the agricultural sector. Farmers increasingly have to provide evidence of their sustainability performance to food processors and government agencies, often in the form of sustainability indicators (e.g., proof of organic farming status, food safety standards such as GlobalGAP, sustainability standards such as 'On the way to planet proof' or 'Bord Bia', Eco-schemes in the Common Agricultural Policy, etc.).

EU member states are not all moving at the same speed on these environmental issues, but in some cases, national-level policy is proceeding at pace. In Ireland, the new Climate Act imposes large greenhouse gas (GHG) emissions reduction targets on agriculture and a requirement to protect biodiversity, while implementing solutions that address climate change impacts. The strong growth in Irish milk production in recent years and the associated increase in dairy-related GHG emissions has caused policymakers to focus

particularly on dairy farming and the need to produce verifiable evidence of progress in emissions mitigation. Similar concerns exist with respect to ammonia emissions, water pollution, and biodiversity, with a clear need for rigorous data, which allows the monitoring of sustainability so that any improvement or deterioration can be identified.

Dutch farmers face similar sustainability obligations and have lobbied for policies based on key performance indicators and mineral balances. Such developments can be considered an indicator of where other member states will need to follow, either led by EU policy or changing societal demands at the national level.

Often such evidence on sustainability performance is required within a system of farm certification. The monitoring and reporting of a farm's environmental and social performance demands some form of farm data management as it is not sufficient to acquire such data by sampling the products of the farm (credence attributes of the products). Neither is it possible for governments to adequately monitor farms using external means, e.g., satellites as the use of antibiotics or pesticides is not observable from the sky.

As farmers derive income (in the form of higher prices, the avoidance of penalties, or receipt of government payments), if their farm performance is compliant with desired standards, data need to be reliable and auditable. Cross-checks, so as to ensure accuracy and completeness of the data, need to be built into the system [1], therefore suggesting that the data required to monitor and report a farm's environmental and social performance should utilize some form of environmental accounting.

In line with this development, the European Commission in its Farm-to-Fork communication [2], has proposed a transformation of its Farm Accountancy Data Network (FADN) into a Farm Sustainability Data Network (FSDN) to improve the monitoring and evaluation of the Common Agricultural Policy (CAP). However, this is just one example where environmental accounting is needed. With the introduction of the new CAP of the anticipated eco-schemes, the EU Sustainable Finance Taxonomy and the demand for environmental and social sustainability indicators from the farm's business partners in the food chain and national policies, nearly all European farmers will face more requests for meaningful and reliable sustainability data.

### 1.1. Accounting as a Basis for Monitoring and Certification

This growing need for farm data could increase the administrative burden on farmers. To address this burden, it is important to identify and make use of the data sources already available and to consider the methods that could be deployed to manage and process the data. For non-farmer outsiders, such as companies in the food chain and government agencies, the farm can look like a black box. It is hard for the outsider to observe what is happening on the farm in terms of key sustainability concerns, such as the use of fertilisers, pesticides, and antibiotics. However, the interactions between a farm and the outside world are well-documented [3]. On most farms, undocumented cash transactions have been replaced with bank transactions and a flow of invoices (and delivery or dispatch documents) for a couple of reasons. Firstly, suppliers of farm inputs (feed, fertiliser, fuel, pesticides, etc.) and food processors or traders of agricultural products want documentation to assist in the administration of their businesses. Secondly, most farmers now have to keep accounts for fiscal purposes (VAT, income tax) and to supply their banks with financial information as part of their financial relationship with such institutions.

Farmers have two well-known accounting methods through which to organize their data: Farm Financial Accounting which is an application of conventional business accounting and Farm Management Information System (FMIS) a software system managing the day-to-day activities of the farm [4]. Farm Financial Accounting uses financial transactions (payment data) to calculate financial statements (for tax purposes and financial management). FMIS is a form of management accounting that developed out of the field records/animal records and registers of inputs and outputs per field and farm activity (crop, type of animals) to guide operational and tactical management decisions. Farm

Financial Accounts focuses on monetary flows (euro amounts) with trade partners and assets, whereas the focus of the FMIS is on volumes and product flows within the farm.

As the FMIS records volume-type data (such as the use of pesticides on certain crops) that are used in operational management, it seems at first sight attractive to use these data as a basis for environmental accounting and the calculation of sustainability indicators. There are however two problems with this approach. First of all, whereas all mid-sized and large farms are obliged to keep financial records for VAT and income tax purposes (and for securing credit from a bank), there are also a lot of farms that do not use an FMIS. It is only the largest farms that have a formal FMIS. Secondly, the FMIS is much more difficult to audit in a certification process. Although nearly all farmers will record their data honestly, mistakes and deliberate misrecordings are not unthinkable, especially if good environmental performance is rewarded with support payments and/or higher output prices (as is the case with eco-schemes, organic products, or products with environmental labels) or bad performance leads to lower payments and/or lower prices.

The integration of environmental and financial accounting makes sustainability indicators auditable [5]. Financial accounting, based on the theory of double-entry accounting, has methods to verify the completeness of its dataset. One is that in modern farming in European societies payments occur by bank transaction. The use of bank account statements guarantees that all payments have been recorded. By linking invoices to these payments, there is some assurance that no invoices have been 'forgotten'; if some invoices were not recorded, which would indicate a lower sustainability level of the farm (e.g., on pesticides), then this would not show up as a deductible cost in VAT and income tax statements. That means that coherence between the sustainability and financial statements is an important aspect of auditing farm data. Of course, these checks are not perfect. Auditors have to check whether or not cash payments were made or if descriptions on invoices are intentionally inaccurate.

The integration of financial and environmental data also has important advantages for decision making by farmers as they are confronted with trade-offs between the financial and environmental aspects of their farm management practices. In reducing the nitrate runoff, a dairy farmer could choose to lower the nitrogen surplus by using less concentrated feed or less fertiliser. Both practices would have an effect on the farm's N-surplus per ha as well as the farm's production costs and revenues.

*1.2. Robotic Accounting*

To control the administrative burden involved in sustainability data provision, it is also important to consider opportunities to reduce the labour required for data management and processing. Robotic process automation (RPA) is aimed at the automatic execution of administrative tasks that reproduce the work that humans do, comparable to robotics in manufacturing. The automation is done with the help of software robots or AI workers that are able to accurately perform repetitive tasks [6]. Applications are found in domains such as purchasing and supply management [7] and also accounting [8]. Robotic process automation (RPA) or robotic accounting is expected to change accounting and auditing significantly [9–11]. RPA software automates the input, processing, and output of data to streamline repetitive, mundane tasks.

The foregoing points prompt the question as to whether robotic accounting has a role in facilitating sustainability monitoring. Robotic accounting could have advantages for financial accounting as such but could bring further benefits given the demand for sustainability monitoring and the advantages of auditing these data by integrating the environmental and financial data. The objective of this paper is to investigate how the monitoring and compliance auditing of farms can be supported with efficient and smart farm data management and processing in the current farm accounting framework.



## 2. Materials and Methods

This paper builds upon the design science paradigm [12–14]. This paradigm seeks to extend the boundaries of human and organizational capabilities by creating new and innovative artifacts. In the design science paradigm, knowledge and understanding of a problem domain and its solution are achieved in the building and application of the designed artifact to solve identified organizational problems. Such artifacts are represented in a structured form that may vary from software, formal logic, and rigorous mathematics to informal natural language descriptions [14]. This paper analyses the problem domain based on a number of use cases. The artifact constructed in this paper consists of the data model, the process model, and the design of a software tool.

To evaluate the potential of robotic accounting for sustainability monitoring in agriculture, we first analyse the extent to which farming-related invoices contain relevant data for sustainability monitoring and environmental accounting. Examples are provided from Irish and Dutch dairy farms. Dairy farming is an important sector in these countries and in both countries sustainability issues have achieved particular prominence in public discourse. Reducing the contribution to climate change is a major challenge. The Irish government [15] announced legally binding national targets for reductions in greenhouse gas emissions, in which agriculture will have to reduce its emissions by at least 22% by 2030 relative to the 2018 level. Dutch dairy farms face a similar challenge [16]. The Nitrates Directive is another issue in both countries [17]. In both countries, the dairy processing industry has introduced strategies for 'more green' dairy production and introduced product labelling.

In the following sections, we develop a data model (entity-type relation diagram) and a process model that support the collection of environmental data from invoices in an efficient way. We analyse a number of use cases to evaluate how the detailed data from invoices can be used in accounting. Furthermore, a process model is developed that supports the integration of bank data and invoice data. These process and data models are crucial for the successful implementation of robotic accounting. Based on these models and the use cases, we present the design of a flexible software tool for robotic accounting.

The paper ends with a discussion on the introduction of robotic accounting in agriculture that also makes some suggestions for further research and closes with conclusions.

## 3. Results

### 3.1. Invoices: A Wealth of Data

Farming-related invoices are nearly always created by farmers' trade partners rather than farmers. Farmers themselves create relatively few invoices relating to the sales of their output. For instance, traditionally it was much more efficient for the dairy cooperative, which could determine the volume and quality of the milk delivered, to issue standardized invoices to its suppliers rather than handle all manner of non-standardized invoices from its supplying farmers. In many countries, it was also the case that documenting the transaction was more important for the cooperative than for the farmer. Only since the introduction of taxation on a real basis (instead of a forfeit basis) and the increasing role of banks, have farmers themselves developed a need to document such transactions [18,19].

An examination of farm invoices from Ireland and the Netherlands shows that invoices contain volume data that can be used to calculate environmental indicators such as pesticide use, mass balances (especially needed in organic farming due to EU Organic Regulations), material balances of N and P, energy use (and production), antibiotics use, etc. A typical example is given in Figure 1, a copy of a real invoice (data relating to the identity of the farm has been redacted) that documents the deliveries and sales by an Irish farmer to his dairy cooperative Dairygold. The invoice provides information on milk deliveries, milk volume, and milk quality in terms of protein, fat, and lactose. It also provides information on the milk price and the amount of money paid to the farmer. This invoice includes bonuses (including in this case even a sustainability bonus) and levies, e.g., for farmer-funded agricultural research. In financial accounting, only the euro amount of money paid is entered into accounting software, which also means that a lot of other useful data that is at

hand is not digitalized and is more or less lost when the invoice is archived. Invoices from other Irish dairy processors, such as Glanbia, Kerry, and Lakelands, contain more or less the same data/information, but the invoices of the various food processors differ strongly in their formats. This hampers the use of optical character recognition (OCR) technology (to scan and interpret these invoices) as discussed in [4].

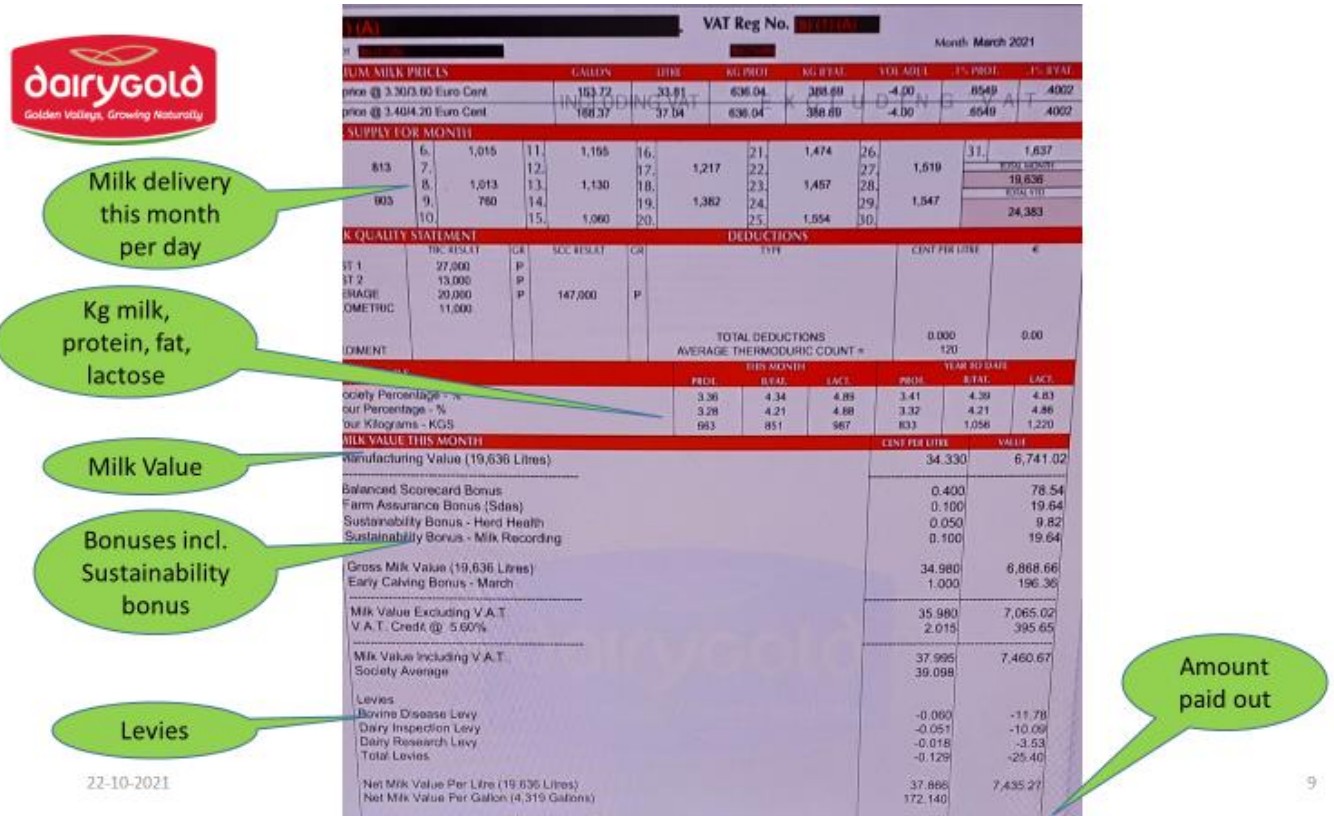

**Figure 1.** Typical invoice that documents milk deliveries and sales by an Irish farmer.

Figure 2 gives a similar example relating to inputs supplied to the farm. Invoices such as this one from the Kerry Trading company provide data at the individual product level, often including a product code designated by the supplier (that could be used to collect even more details on the nutritional or chemical composition of those products by contacting the suppliers that are large companies who sell to a large number of farms). In total the invoice has 18 product lines (linked to the dairy operation in the first 11 and others in the second 7) with quantities, the unit in which quantities are measured, the price per unit, and the total costs. Such detail on invoices seems to be a normal business practice; Glanbia Trading provides the same information and, as we will see below, Dutch companies also do this.

Irish and Dutch invoices for dairy farms proved to be very comparable concerning the detailed data provided. Some Dutch examples are reported in the next section as they are used to illustrate aspects of robotic accounting.

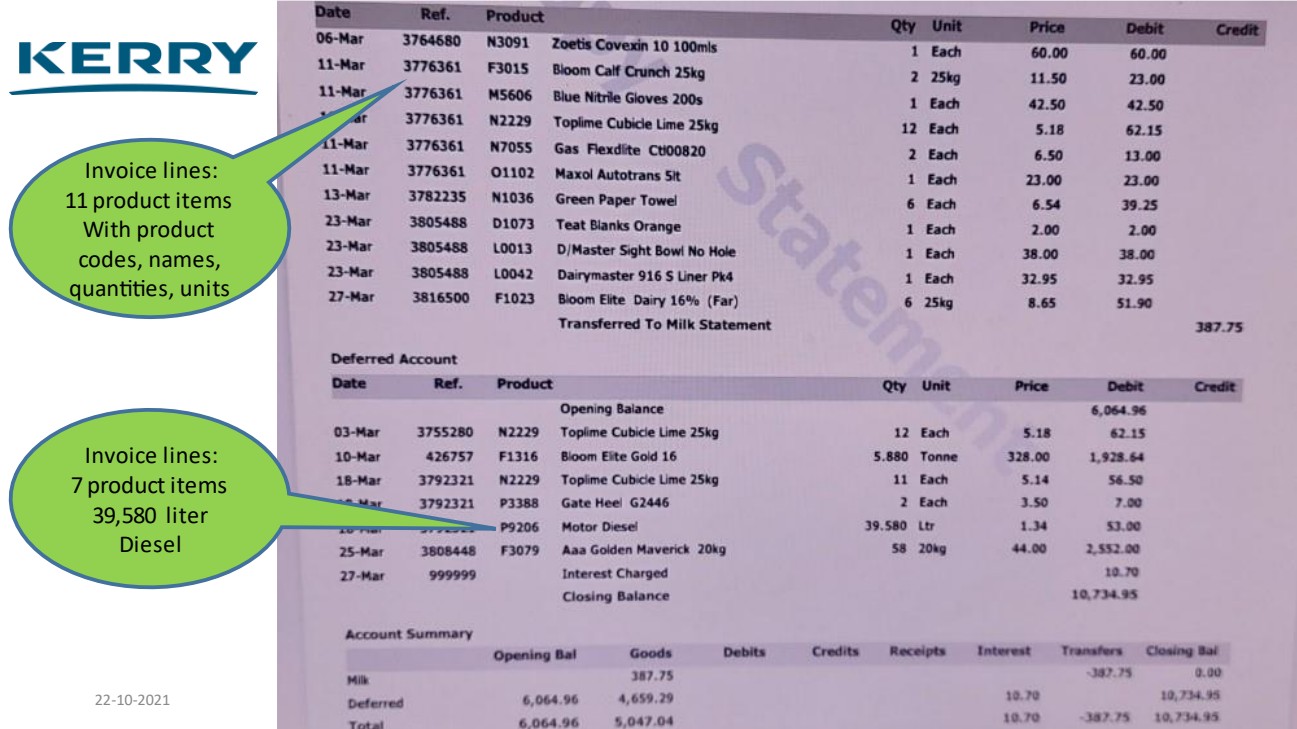

**Figure 2.** Typical invoice (body of the invoice only) on inputs supplied to an Irish farm by input supplier Kerry Trading.

### 3.2. The Use of Invoice Data in Robotic Accounting

Based on the data in the invoices, we analyse a number of use cases to evaluate whether the detailed data from invoices can be used in accounting. Currently, farm financial accounts mainly use the amount paid and code that, based on the invoice, for input items or sales of output. In some cases, the amount paid on an invoice is split into different types of inputs (e.g., the cost of motor diesel and the cost of various dairy supplies in Figure 2, which are recorded as individual types of costs).

To investigate the potential of robotic accounting for recording the details of such invoices, a data model (entity-type relation diagram [20]) was created. This basis of robotic accounting can be used to create a database of coded transactions of a farm with its suppliers and clients. The model is relatively simple (Figure 3). A farm receives invoices from suppliers (suppliers of invoices include the businesses that purchase the products of the farm). These invoices contain one or more transactions (lines of an invoice) for products that are typical for the supplier. Invoices can be matched with (and sometimes created from) bank payments. Reports with indicators are aggregations of coded transactions.

An essential requirement for robotic accounting is a classification table that links the names of the products in the transactions of suppliers as well as the descriptions in the bank transactions to standardized accounting names and ledger codes in the accounting software (or in the FADN). Figure 4 provides an example of this table.

To illustrate this form of robotic accounting, we analyse a number of use cases describing the ways in which a user interacts with an accounting system. We describe the use cases based on the dialogue box and the software processes initiated or linked to the dialogue box. The first use case illustrates how robotic accounting can help the efficient recording of environmental data by supporting coding with an automatic allocation of discounts to products and linking of product names to accounting codes, using an example based on a Dutch invoice for feed supplies. The invoice contains detailed data on individual types of feed and volumes that can be recorded. The invoice also states that the feed supplier calculated four types of discounts on the transactions: advance ordering, ordering a full lorry, cumulative annual ordering, and direct payment based on a bank authorization.

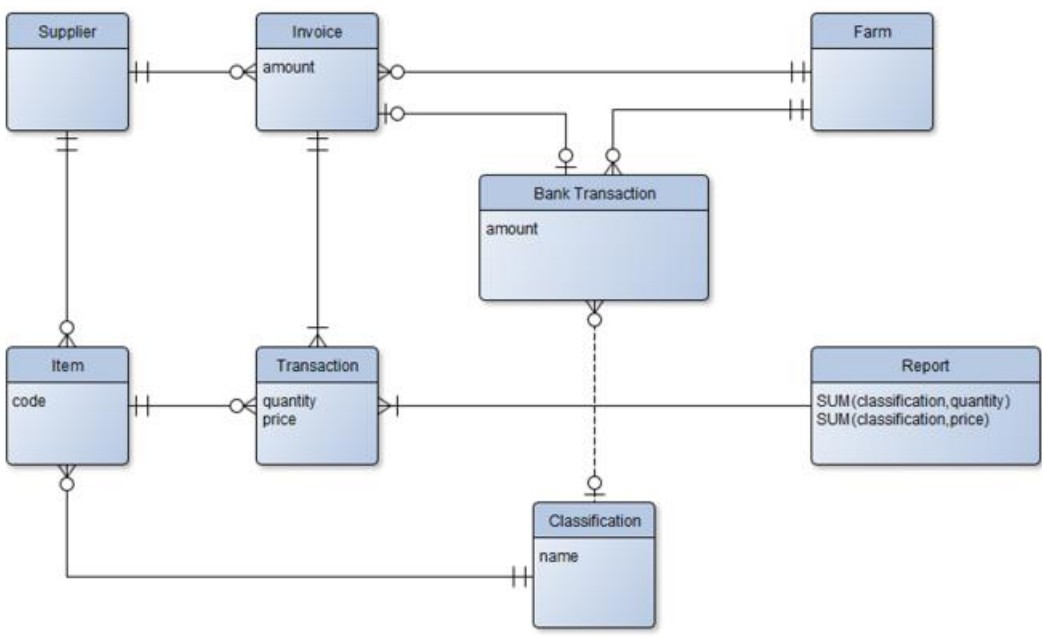

**Figure 3.** Data model for robotic accounting.

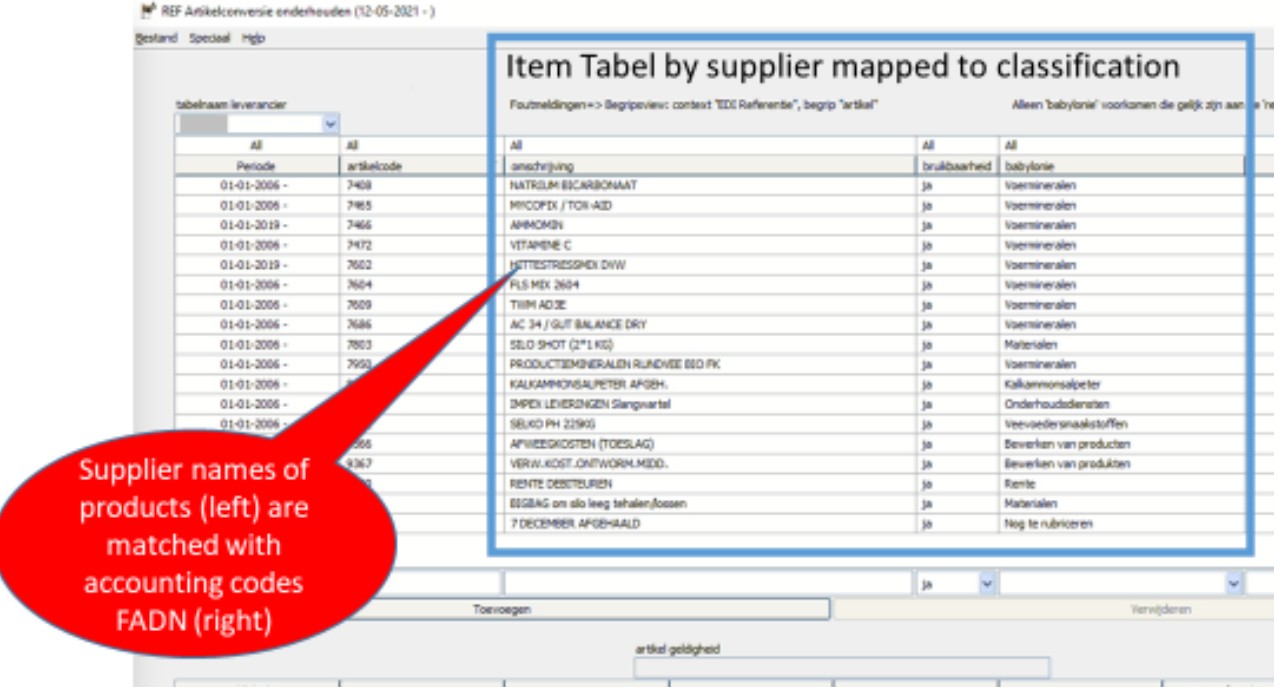

**Figure 4.** Example of the classification table in robotic accounting (SITRA).

The right panel of Figure 5 contains a dialogue box showing how supplementary data from the invoice can be entered in tandem with the regular financial data. At the top of the dialogue box, the total amount of the invoice is entered, and the accountant (or farmer) can add the product names (or numbers) from the invoice. The entry of product names can be supported by the software recognizing the product from its database and linking it to an FADN accounting code—in this case feed. The accountant can enter the volume in kg and the value of the transaction. Subsequently, the software can calculate a unit price that is available for a visual check and a check by the software to see if it is within a predefined range. Based on the (standard) type of feed, the feed is also allocated as a cost to the dairy operation (in this example, for typical pig feed it would be allocated to the pig enterprise).

Based on the information provided by the supplier on the types of feed (or on the invoice data), the amount of phosphate (P) and nitrogen (N) in the feed can also be calculated, which is of relevance in order to calculate material balances for N and P. The software also automatically allocates the discounts to the different types of feed and can handle data entry including or excluding VAT.

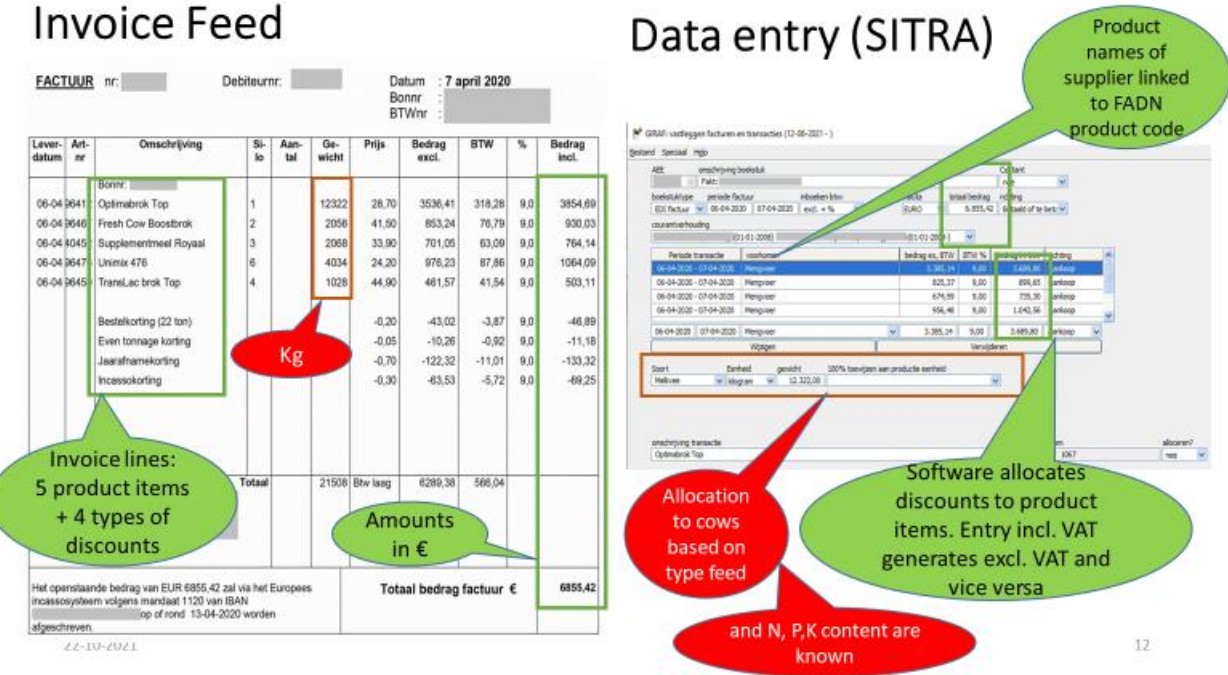

**Figure 5.** Use case on the invoice of a feed supplier (left panel) and the data entry dialog box and processing SITRA (right panel)—in Dutch.

Figure 6 provides a similar use case in which an (organic) dairy farmer bought inputs used on his grassland. In this example, the invoice states the volume of one product in terms of the number of cans purchased. Based on data from the supplier (or an earlier entry on one of the farms in the accounting software), the software converts the cans to litres at the time of data entry. Also at this point, unit prices are calculated as a validity check. The figure also shows how a discount for prompt payment is allocated to the different products (alternatively the robotic accounting algorithm could allocate this discount to interest income).

These use cases clearly show how volume data included in invoices can be recorded as part of an accounting process originally designed to gather data for financial accounting purposes. Dedicated software helps to minimise data entry by hand and reduces the number of data entry errors. Nevertheless, when recording additional data (such as volume data) there is additional work involved compared to the conventional financial accounting process.

In the next use case, we consider a situation where the additional work required to record volume data or other ancillary data would no longer be required in a process of robotic accounting once the invoices have been made digitally available using digital standards such as UBL, XML, UNCEFACT, etc. Such a digital data exchange has the advantage that farmers or their accountants do not have to type the financial data into a database for VAT and income tax accounting (financial accounting) and that volume data, as well as environmental data (sustainability indicators), could be generated for management dashboards at little or no additional cost (management accounting). These metrics can then be supplied to food chain partners, certification bodies, or the government.

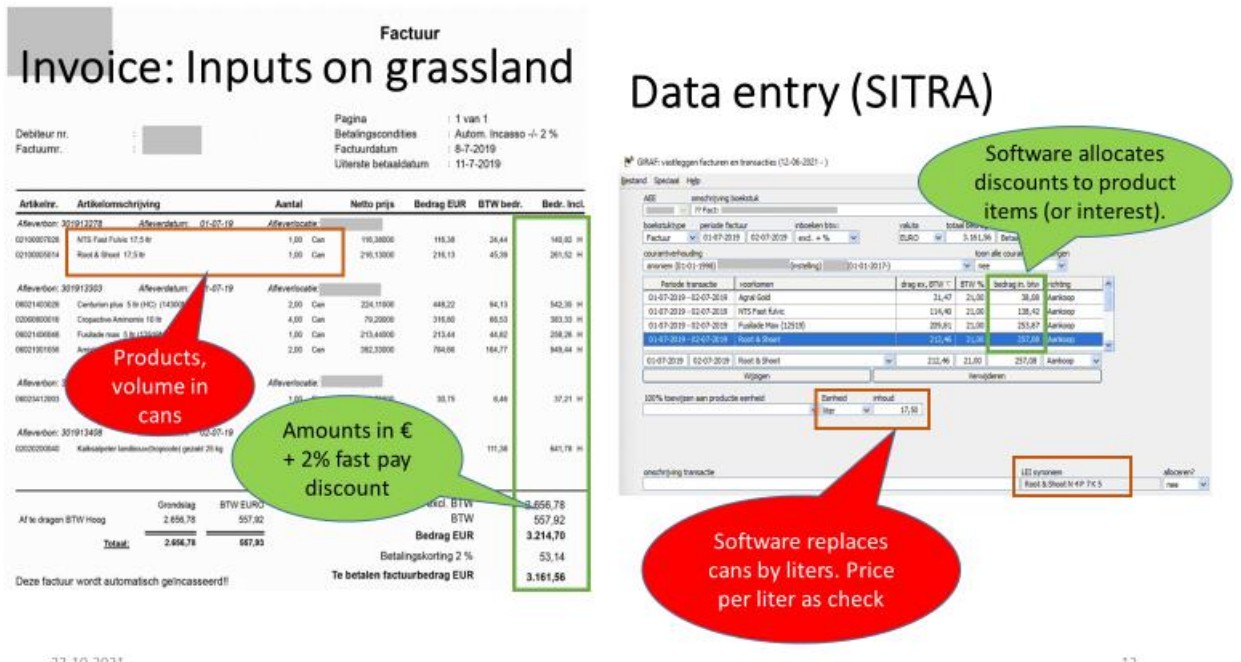

**Figure 6.** Use case on the invoice of a supplier of inputs (left panel) and the data entry dialogue box and processing logic (right panel)—in Dutch.

Such a procedure based on the data model provided above (Figure 3) is feasible for some invoices, especially those for milk and feed, in the Netherlands. Figure 7 shows an invoice that is very comparable to the Irish example in Figure 1. This invoice is not only provided on paper or in a pdf format but also in an XML format. That makes it possible to automatically load the invoice in the accounting software and show it in a dialog box, dispensing with the requirement for manual data entry.

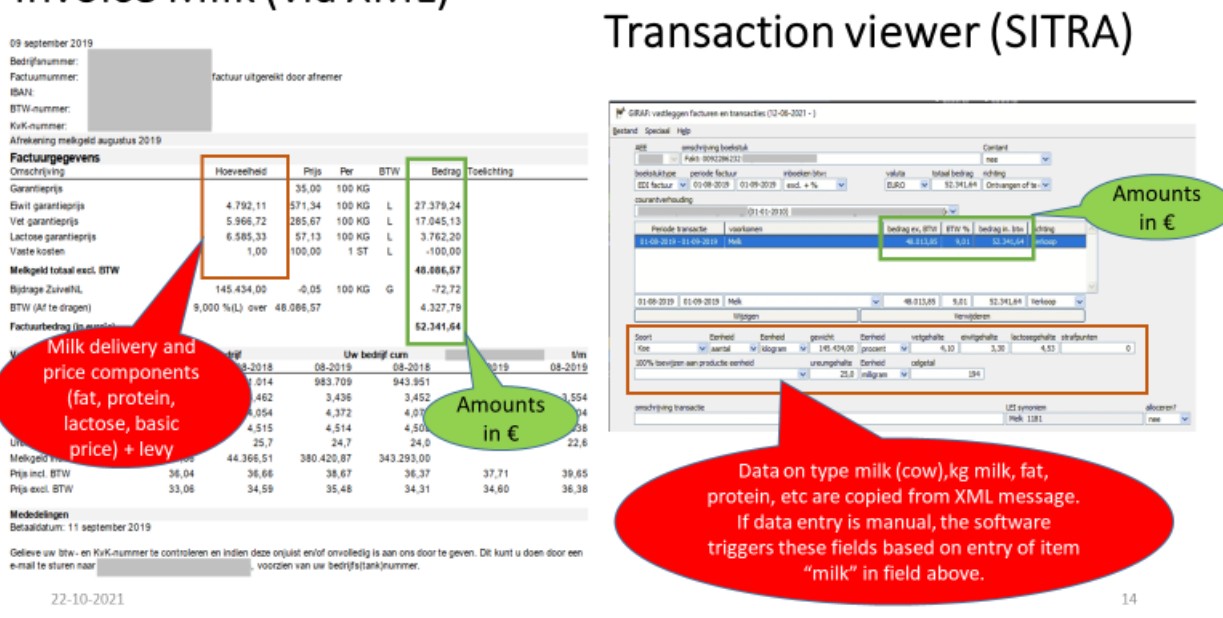

**Figure 7.** Use case on invoice for milk sales by a Dutch farm supplied by the dairy cooperative on paper/pdf and in XML format (left panel) and a view of the automatic data entry in accounting software SITRA (right panel).

This use case, relating to the milk example (Figure 7), could easily be extended to other invoices that are supplied in XML or another computer-readable digital format such as UBL, XML, UNCEFACT, etc. In future, if all invoices were made available in such a format, the cost incurred in undertaking both environmental and financial accounting would be greatly reduced and could even become marginal, as the costs of maintaining the algorithms could be shared by other users of the software platform.

The maintenance of the classification table (Figure 4) and related tables (e.g., that provide checkpoints for unit prices or help to convert volume units such as cans into liters or kilograms) is preferably done centrally in the accounting software platform, making an up-to-date classification table accessible to a broader group of users, which is based on the suggestions/requirements of those doing the data entry (farmers and accountants). Centralising the maintenance of the classification table not only shares the maintenance costs across all users but it also supports auditability (with a classification table common to all users it is difficult to code transactions in a non-standard way, e.g., users must record diesel as fuel and thus it is included in a $CO_2$ indicator calculation, and users are unable to omit the diesel by, for example, coding it under 'general costs') and it guarantees comparability between farms for benchmarking purposes.

### 3.3. The Role of Bank Data in Robotic Accounting

In principle, the accounting process based on (paid) invoices, as described above, is enough to create and integrate financial and environmental accounting. This is particularly the case if farm-level bookkeeping is well-organized and there is a low risk of fraud, as the income of the farm is independent of the sustainability indicator values calculated from the data. This is, for instance, the case in the proposed Farm Sustainability Data Network and many of the current private product labelling initiatives.

However, as sustainability indicators become more important (e.g., for the Eco-schemes in the CAP), it makes sense to make the accounting process auditable. This can be done by linking the invoice data with bank payment data.

Bank data are digitally available to farmers (and with the authorisation of the farmer for their accountants or software suppliers) under the PSD2 banking regulation, which obliges banks to make payment data digitally available to the owner of the bank account upon request. This implies that the linkage with invoices has low associated costs while providing several advantages.

Linking invoices with bank data provides an advantage with respect to completeness, reliability, and efficiency; it guarantees that all invoices (paid) have been taken into account and that invoices have not been lost (by accident or on purpose). Were an individual to buy inputs with cash (paid out of consumer household expenses if not recorded as a business expense) and if the invoice for inputs was deliberately not recorded, the cost is then omitted from the individual's financial records and would not be deductible as a cost for VAT and income tax purposes [5]. This also increases the reliability of the accounts as it makes the volume data auditable; not only are the invoices complete but also the data entry method and a check on unit prices create confidence in the volumes recorded. Third, it makes financial accounting more efficient as some payment data can be coded directly (that is, a dummy invoice can be created) as the payment has all the information needed for the accounts, for example, payments to a national telecommunication company (=communication costs) or a credit card payment for household expenses.

To investigate the potential of robotic accounting for the accounting process described in the previous sections in which we combine financial and environmental accounting as much as possible based on digital invoices and bank data with algorithms to code the data, a simple process model that describes the workflow in robotic accounting for farm financial and environmental data management has been developed (Figure 8). It is complementary to the data model in Figure 3. Invoices arrive in paper (pdf) and digital forms. These are entered into the data store and coded. Coding is a form of classification based on specified rules. Digital bank data is entered in the data store, coded (where useful), and matched

with the invoices. This leads to a data store with coded and auditable transactions (and payments). These transactions are the basis for the reporting of financial, environmental, and social indicators (sustainability indicators or key performance indicators). For accrual accounting, an additional workflow is needed to record data from stocktaking on a balance sheet date that closes the yearly or quarterly accounts.

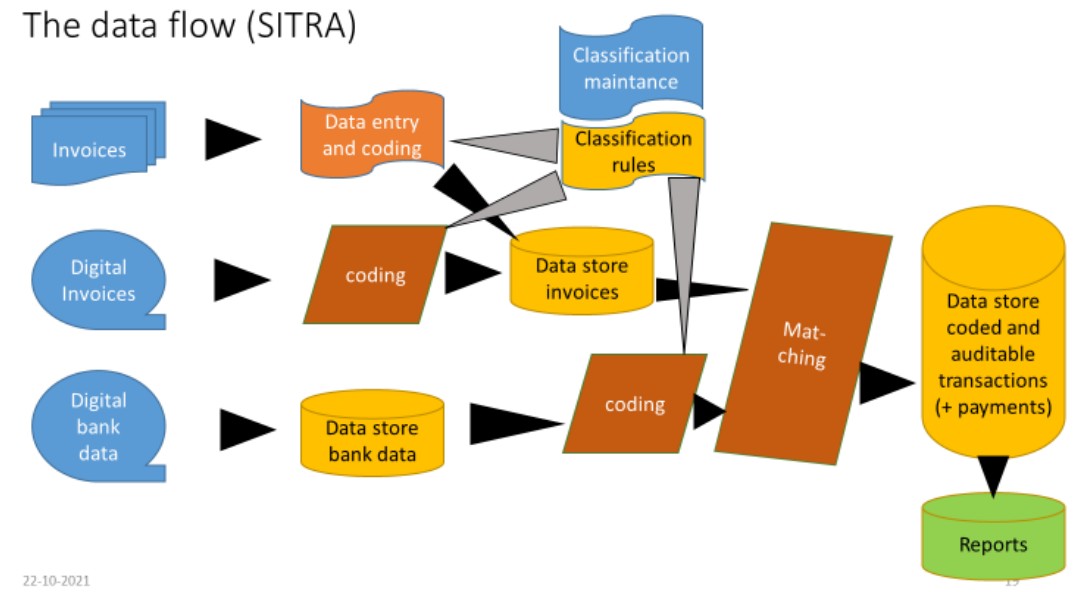

**Figure 8.** Process model for the workflow in robotic accounting.

A use case describing this approach is given in Figure 9. Dialogue boxes show bank transactions and invoices. The software can match bank transactions and invoices based on the amount of the payment or the name/bank account of the trading partner of the farm. Features in the software cater for payments that relate to more than one invoice or one invoice that is paid in instalments.

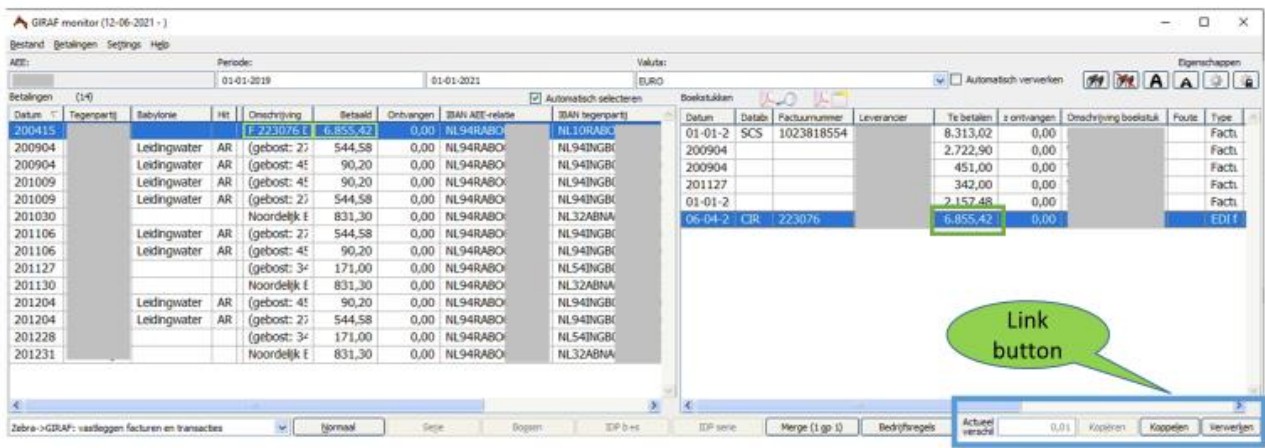

**Figure 9.** Matching invoice data with digital bank data.

### 3.4. Design of a Flexible Robotic Accounting Tool

Robotic accounting, as described in the previous part of this paper, would benefit from a flexible IT tool for two reasons. First of all, it is unlikely that all suppliers of invoices can simultaneously commit to a transition where they then provide these documents in a digital format. This implies that in the future as this transition to digital invoices progresses, workflows for manual data entry and for digital entry have to be maintained and added. Secondly, as sustainability indicators change with the emergence of new environmental or

social priorities and as data suppliers progress to provide additional data on their invoices, this also requires flexibility in the capacity of the software as workflow procedures have to be updated.

It is very attractive to design software that gives staff using the software for robotic accounting the ability to set up procedures in which new types of data can be acquired or new indicators calculated, obviating the need to resort to programmers to revise the software code [21]. A design for such software is shown in Figure 10. It shows a data model that supports a workflow procedure that can be broken down into several acts that use certain software system components in a preset configuration. For an act of a working procedure, an instruction can be created that describes to the user how the items of the database can be created, read, updated, or deleted for that step of the procedure. Such items (entities and their aspects) have data values.

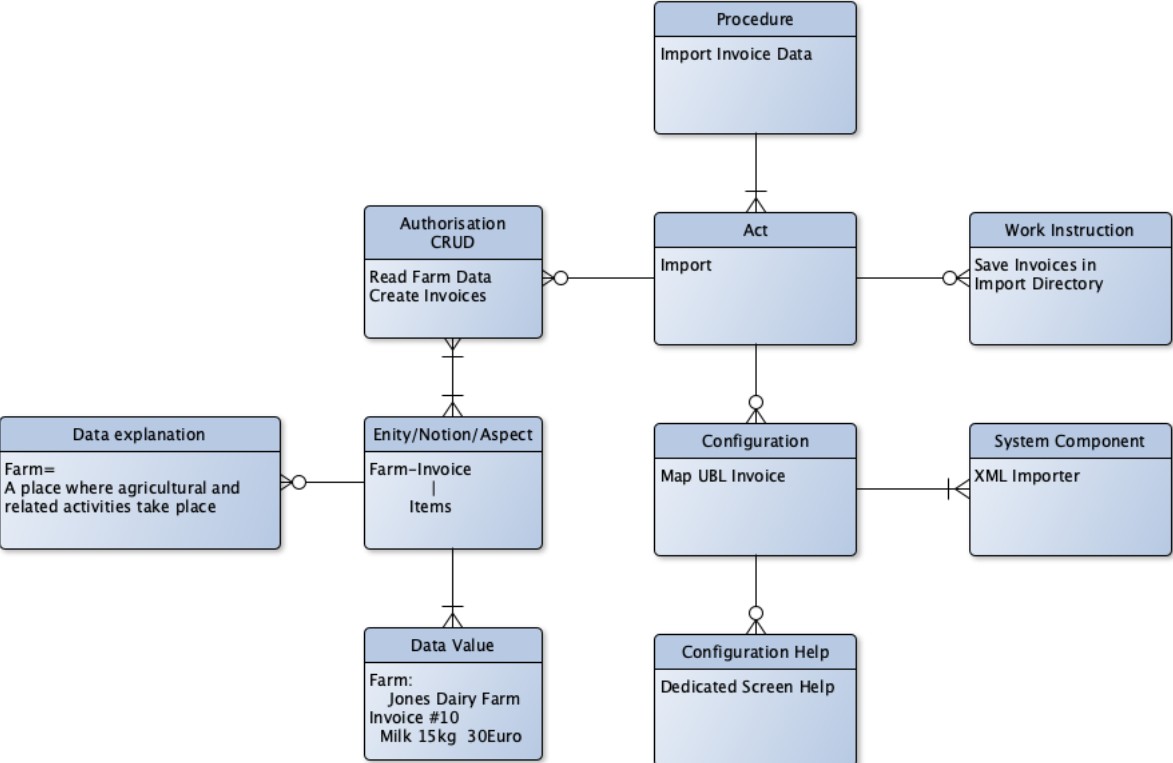

**Figure 10.** Simplified design for a flexible robotic accounting tool (ARTIS).

An illustration would be the import into the database of an invoice from a dairy company. Such a procedure would have several acts (such as running an API to connect to the database of the dairy company every evening, importing the data and storing it in the database, etc.) with a work instruction. In such an act, the software would use a particular system component (e.g., an XML importer) with a particular configuration (e.g., version of the software). In the procedure, the software is authorized to read the identifier data of the farm (e.g., its unique client number with the dairy company) and create an invoice in the database with several aspects of the data (e.g., number of the invoice, milk in kg, and sales in euro).

In this way, the data entry procedure with (relevant environmental) volume data that triggers fields for certain data elements based on the entry of a type of product (item) is very flexible, as new sustainability objectives, which require the recording of additional data, can emerge and can be incorporated. Data are checked by rules for data items on their relevance (for which items are supplementary volume data needed?), integrity (are data within a certain confidence interval?), and actuality (available in time?). The outcomes of these checks lead to a workflow for the farmer/accountant in the form of a to-do list.

## 4. Discussion

In this paper, we have shown how invoices contain a wealth of information that can be used to calculate sustainability indicators in a reliable way for all farms that are integrated into the market economy. The transition to a real-time economy is currently underway and in light of this, the introduction of robotic accounting would deliver benefits for the financial accounting process, primarily reducing the cost of data entry. However, robotic accounting can do more than that, as it efficiently addresses the increasing amount of data required to measure the environmental performance of farms for sustainability monitoring purposes. Furthermore, robotic accounting can facilitate the auditing of these data by integrating environmental and financial data. The approach depends on the availability of suitably designed invoices in standardized formats (and or other documents such as delivery notes) that farmers receive from the suppliers of their inputs and buyers of their outputs. This represents an obstacle to be overcome and requires collaborative efforts on the part of those providing such invoices.

This use of robotic accounting to match bank data and invoice data leads to a fully automated system of cash accounting. For accrual accounting, inventories have to be recorded; some inputs such as feed or fertiliser are invoiced and delivered but are not used in the accounting year. A manual process that supplements robotic accounting will be required to ensure that usage is accurately recorded making allowance for changes in inventories.

There are some cases where the reduction of the administrative burden through the use of robotic accounting will never be fully realized. An example is a situation where a farm sells some of its products through cash-based transactions, e.g., in its own farm shop. Such farm shops do not have the advanced cash registers of the big retailers, which means that sales per product are poorly documented. A farmer will have to document the movement of the products from the farmyard/warehouse to the farm shop (in transfer prices and in consumer sales prices to simplify the administration of the shop). Some farmers send invoices themselves. We came across an example in which a group of farmers exchange their products once a week to supply each other's farm shops to complete the assortment of items for sale. However, it turned out that they also invoice each other, which implies that such situations can be addressed by our proposed solution. In some cases, farmers acquire inputs more or less free of charge. We came across an example where an organic farmer received mowed grass (of bad quality) from a nature reserve free of charge for his barn. However, the contractor who collected the grass for the farmer did send an invoice for his work, specifying the number of bales (that have a standard size and weight). In a flexible accounting tool, such as the one we described, workflow procedures could be added to capture such cases with some manual intervention where needed. The reliability of such transactions is lower in any accounting system and will not be solved by implementing robotic accounting to link invoices and environmental data to bank data.

The approach described in this paper to calculate sustainability indicators and mass balances with the help of invoices could be applied directly by companies providing farm financial accounting software or farm information systems to farmers as well as by accounting offices, banks, advisory services, and (cooperative) food processors. National government ministries that want to support farmers in reducing the administrative burden involved in monitoring farm sustainability would have an important coordinating role. It could also be applied by FADN partners who would like to progress towards a Farm Sustainability Data Network [22].

The development of robotic accounting and its adoption by users now requires the attention of both companies in the food chain and national governments. In this regard, it is useful to reflect on some history in the Netherlands, where digital invoices were introduced as early as the 1990s. The potential benefit that would be derived from the widespread adoption of digital invoices remained ignored and unexploited by many companies in the Netherlands that send invoices to farmers. This suggests some market failure due to the fact that, in this case, the adoption process works best if all partners in the food chain make

the transition to digital invoices collectively. Such a transition probably asks for a common awareness of the need for environmental accounting and the incorporation of options in the software that food chain companies use that would allow digital invoices to be created without too much effort.

Companies in the food chain have to incur some costs in the introduction of digital invoices for the benefit of farmers and their accountants. Governments requiring farmers to improve their environmental performance and provide sustainability data should seriously consider making digital invoices with environmentally relevant details a standard for businesses that transact with farmers. For instance, the Hungarian government has obliged companies (in all sectors) to use digital invoices although the motivation for this was different (checks on tax evasion). Further investigation of the costs and benefits of robotic accounting could represent a follow-up topic in this research area.

In the approach we have described, we have assumed that buying inputs is equivalent to their subsequent use and equally that sales are equivalent to output. This is not necessarily the case as farmers can temporarily hold stocks of inputs until they are required for use. Equally, farm outputs can be stored on the farm until their moment of sale. In such cases, it would make sense, also for financial accounting purposes, to add a workflow procedure in which such inventories are entered into the database to calculate indicators on an accrual basis instead of a cashflow basis. For those farms that use an FMIS, data on the usage of inputs could be linked to the data set of the coded transactions that our approach generates from the invoices. This has the advantage that more detailed information on the use of the inputs is generated, including an allocation to crops, fields, and (individual) animals. Such a link would also ensure that data is auditable in a certification process, addressing and solving the problem of the potential unreliability of current FMIS data. The same benefits can be derived from sensor data which are often integrated into an FMIS. In this way, a unique sustainability dashboard for farmers would become available. More work on that integration is foreseen in the MEF4CAP project.

Future research, as indicated above, could take a range of different directions. The design could be tested with farmers in a real setting with mock-ups of the digital invoices. The approach could also be extended by integrating data from farm management information systems and sensors. Research on the costs and benefits of such accounting systems and measuring the reduction in the administrative burden that could be achieved, are other possible areas for investigation. How a collective action should be organised to overcome problems in the adoption of robotic accounting and especially the digitalisation of invoices, seems to be a country-specific issue that depends on the national institutional setting but could provide a topic for some interesting case studies. Once data are available from the system we designed, a whole field of research on environmental management in agriculture and the relevance of KPI opens up.

## 5. Conclusions

In applying the design science paradigm this paper designs new and innovative artifacts to support future sustainability monitoring. Our analysis of Irish and Dutch invoices has led to the specification of a data and process model, which we have demonstrated in a number of use cases. These show how it is possible to process specific invoices to provide integrated financial and environmental accounting.

This integrated approach has three advantages. Firstly, sustainability indicators could be generated for all farms integrated into the market economy and especially those that are already obliged to keep books for income tax or VAT purposes. This encompasses a much wider group of farmers than the number using a Farm Management Information System and would thus reduce the administrative burden for many farmers. Under the proposed approach, farmers would not be obliged to adopt an FMIS, nor would they have to enter their farm data on a food processor's website.

Secondly, the approach provides farmers with integrated data that support decision making to weigh up the trade-offs between the financial and environmental aspects of their

management. The integration of financial and environmental data could also facilitate the auditing of such data for accuracy.

This integration could make data available on such issues as the use of antibiotics, pesticides, and fertilisers, all of which are key priorities in the EU's Farm to Fork initiative. It makes it possible to calculate farm-gate material balances for nitrogen and phosphate, as well as mass balances that are used in the certification of organic farms. Energy inputs could be used to calculate a farm's $CO_2$ emissions.

Farmers could use these sustainability indicators, calculated from environmental and financial accounting, in their farm management decision making and, if required, could also share the data with food chain programs, government organizations (paying agencies for eco-schemes, organic certification), and the FSDN. The FSDN could use this for monitoring and reporting purposes (see [23–25] and as an early example [26]).

Thirdly, robotic accounting would greatly reduce the cost of sustainability monitoring. It provides a solid basis for environmental accounting and also provides benefits to financial accounting. Such a transition to robotic accounting would require that farms receive their invoices in a digital (UBL) format (whereas other more traditional forms of invoice provision could persist if necessary) as this would lead to reduced accounting costs and can at the same time also provide farms with a dashboard of sustainability indicators. Once invoices are digitalised, accounting costs drop and the administrative burden associated with environmental accounting diminishes due to the low marginal cost of data management.

As the digitalisation of invoices is unlikely to happen overnight (unless governments oblige this), the sustainability indicators and the IT tools required to capture data relating to them will continue to evolve alongside the transition to digital invoicing. Therefore, it would be attractive to have a flexible IT tool, capable of being tailored by the user to capture whatever additional data is deemed necessary. This solution is now available for the Dutch FADN. The next step is to see if this IT tool can be further enhanced to populate individual farm dashboards.

**Author Contributions:** K.P. contributed to the conceptualization, methodology, formal analysis, and writing of the paper. H.V., E.D. and T.D. contributed to the methodology, the formal analysis, the writing and review, and editing. R.v.D. and N.d.G. contributed to the software, data curation, and data visualization. All authors have read and agreed to the published version of the manuscript.

**Funding:** This project has received funding from the European Union's Horizon 2020 research and innovation programme under grant agreement No. 101000662. The content of this publication exclusively reflects the author's view and the Research Executive Agency and the Commission are not responsible for any use that may be made of the information it contains.

**Institutional Review Board Statement:** Not applicable.

**Informed Consent Statement:** Not applicable.

**Data Availability Statement:** Not applicable.

**Acknowledgments:** This paper has been written in the context of the European MEF4CAP project and builds upon a case that aims to realize a System for Information Transfer to Reduce Administrative Burdens (SITRA) in the agrifood sector [4].

**Conflicts of Interest:** The authors declare no conflict of interest.

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
