# Peer review of "Sustainability Monitoring with Robotic Accounting—Integration of Financial and Environmental Farm Data"

_sustainability, doi:10.3390/su14116756_

Round 1

Reviewer 1 Report

The topic of the article is modern and deals with an important aspect of sustainability reporting. The data that can be collected from financial accounting documents are well illustrated by examples.

The claim that financial accounting is based on pay-as-you-go accounting raises some questions. Although the article also mentions that transactions and their accounting are controlled through the debit credit system, the article states that it is important to link the environmental data collection system to the payments.

Thus, the article could make it clearer why digital linking of payments / receipts is important. This is a feature that most financial accounting software today offers in collaboration with banks. So the question is why it should be implemented, not so much as how it could be technically implemented.

Thus, the role of the bank link in this article could be considered.

The article also points out that the corresponding digitization and automatic financial accounting systems have been available for quite some time (approximately 25 years) and the question is rather why they are not used. The transition to a real-time economy is currently under way, and in this light the integration of databases, including the integration of environmental data databases with financial accounting, is certainly on the agenda in all countries. However, as pointed out in this article, the obstacle is to use these existing possibilities, ie the social rather than the technical side, to which the article briefly refers.

It may be useful to highlight more clearly the problems associated with use and the opportunities and obstacles to overcome them.

Reviewer 2 Report

Thank you for the opportunity to review this interesting article. After reading it I found the following aspects related to:   1. Abstract. The authors specify the objective of the research, the results and the conclusions found. In my opinion the abstract is a bit long, there may be a possibility that it can be slightly reduced!   2. Introduction. This section contains a part of the Specialized Literature but is not delimited with special subchapters for the notions treated so that the readers can decipher more easily the notions and concepts used by the authors. I suggest a subchapter reconfiguration of the Introduction section and highlighting the essential issues!   3. Materials and methods. Simple and comprehensive description!   4. Results. Very well structured section with interesting results, especially the model on page 7.   5. Discussion. A section that clarifies what the authors found.   6. Conclusions. This section lacks the limitations of the study conducted by the authors and the directions of future research, which I recommend you add!

Reviewer 3 Report

I suggest to improve the theoretical framework and the hypotheses development

Reviewer 4 Report

The objective of this paper is to investigate how monitoring and compliance auditing of farms can be supported with robotic accounting for efficient and smart farm data management based on existing methods of farm financial accounting.

The paper is well written. Although the work is very descriptive, I understand that it makes an interesting contribution from the point of view of farm data management in order to integrate financial accounting and environmental accounting. The results are useful to improve the sustainability of agricultural producers.

The approach that is exposed based on the data capture of paid invoices is highly biased by the cash criterion. Therefore, the authors need to explain in more detail whether rototized accounting allows the accrual principle to be properly applied. For example, the purchase of feed and fodder may not coincide with annual consumption, so the environmental impacts would not be well reflected through purchases.

If the invoices paid by the bank give robotic accounting reliability, how is this reliability guaranteed in operations that do not involve payment?

How do you contemplate service subcontracting operations where the invoice does not include physical amounts to estimate environmental impacts?

It should be explained in more detail which agents should be responsible for robotic accounting: producers, cooperatives, advisors, etc.

The collaboration processes between different sectoral agents should be explained in more detail so that robotic accounting properly reflects the economic and environmental reality of the farms.

Round 2

Reviewer 4 Report

The new version has improved the work